# Molecular Evolution of SNAREs in *Vitis vinifera* and Expression Analysis under Phytohormones and Abiotic Stress

**DOI:** 10.3390/ijms25115984

**Published:** 2024-05-30

**Authors:** Bao-zhen Zeng, Xue-ting Zhou, Hui-min Gou, Li-li Che, Shi-xiong Lu, Juan-bo Yang, Yong-juan Cheng, Guo-ping Liang, Juan Mao

**Affiliations:** College of Horticulture, Gansu Agricultural University, Lanzhou 730070, China; 18394426316@163.com (B.-z.Z.); 15101219013@163.com (X.-t.Z.); ghm1648885861@163.com (H.-m.G.); 18209325789@163.com (L.-l.C.); 18893912407@163.com (S.-x.L.); 15609437995@163.com (J.-b.Y.); 18394512517@163.com (Y.-j.C.); lianggp@gsau.edu.cn (G.-p.L.)

**Keywords:** SNARE gene family, abiotic stress, qRT-PCR, grapevine, subcellular localization

## Abstract

SNARE proteins (soluble N-ethylmaleimide-sensitive factor attachment protein receptors) play a key role in mediating a variety of plant biological processes. Currently, the function of the SNARE gene family in phytohormonal and abiotic stress treatments in grapevine is currently unknown, making it worthwhile to characterize and analyze the function and expression of this family in grapevine. In the present study, 52 *VvSNARE* genes were identified and predominantly distributed on 18 chromosomes. Secondary structures showed that the *VvSNARE* genes family irregular random coils and α-helices. The promoter regions of the *VvSNARE* genes were enriched for light-, abiotic-stress-, and hormone-responsive elements. Intraspecific collinearity analysis identified 10 pairs collinear genes within the *VvSNARE* family and unveiled a greater number of collinear genes between grapevine and apple, as well as *Arabidopsis thaliana*, but less associations with *Oryza sativa*. Quantitative real-time PCR (qRT-PCR) analyses showed that the *VvSNARE* genes have response to treatments with ABA, NaCl, PEG, and 4 °C. Notably, *VvSNARE2*, *VvSNARE14*, *VvSNARE15*, and *VvSNARE17* showed up-regulation in response to ABA treatment. *VvSNARE2*, *VvSNARE15*, *VvSNARE18*, *VvSNARE19*, *VvSNARE20*, *VvSNARE24*, *VvSNARE25,* and *VvSNARE29* exhibited significant up-regulation when exposed to NaCl treatment. The PEG treatment led to significant down-regulation of *VvSNARE1*, *VvSNARE8*, *VvSNARE23*, *VvSNARE25*, *VvSNARE26*, *VvSNARE31,* and *VvSNARE49* gene expression. The expression levels of *VvSNARE37*, *VvSNARE44,* and *VvSNARE46* were significantly enhanced after exposure to 4 °C treatment. Furthermore, subcellular localization assays certified that *VvSNARE37*, *VvSNARE44*, and *VvSNARE46* were specifically localized at the cell membrane. Overall, this study showed the critical role of the *VvSNARE* genes family in the abiotic stress response of grapevines, thereby providing novel candidate genes such as *VvSNARE37*, *VvSNARE44*, and *VvSNARE46* for further exploration in grapevine stress tolerance research.

## 1. Introduction

The endomembrane system in eukaryotic cells is comprised of an array of membrane structures, such as the nuclear and mitochondrial membranes, and engages in the exchange of substances via vesicular transport mechanisms [1]. Vesicular transport represents a multifaceted process encompassing vesicle outgrowth, transportation, tethering, and fusion [2]. Notably, the fusion of the vesicular membrane with the target membrane stands out as a critical aspect [3], which necessitates the mediation by the soluble N-ethylmaleimide-sensitive factor attachment protein receptor (*SNARE*) [4]. *SNARE* genes facilitate the translocation of substances and information primarily via the formation of complexes and mediate the fusion of disparate organelles or membranes on cell membranes [5,6]. Novick et al. [7] conducted screenings for temperature-sensitive mutants and identified genes regulating membrane fusion, including *SEC17*, *SEC18*, *SEC20*, and *SEC22*. The *SEC* gene, denoting selenium-binding protein, encodes a category of selenium-associated proteins that are integral in storing and transporting selenium within cells. SNARE protein, as a key mediator in eukaryotic biofilm fusion process, has a low number of molecules, contains 100–300 amino acid residues and a domain containing about 60 residues [8,9], and shows a high degree of structural conservation in plant species [10,11]. In particular, in eukaryotic cells with complex tissues, the fusion between vesicles and target membranes is promoted through the formation of SNARE complexes [12].

Prior research has demonstrated that stress environments significantly influence the expression levels of *SNARE* genes and the functionality of SNARE proteins. Environmental stressors such as drought, osmotic stress, high salinity, and ABA stress can induce alterations in *SNARE* gene expression, thereby impacting the synthesis and function of SNARE proteins [13,14]. In response to stressful environments, the expression of *SNARE* genes might be modulated by upstream signaling pathways. As an illustration, the activation and engagement of transcription factor binding sites on promoters facilitates the transcription of *SNARE* genes [15]. Moreover, various studies indicate that stress can modulate *SNARE* gene expression through alterations in DNA methylation and histone modifications [16]. The exposure to low temperature can result in structural alterations of SNARE proteins, consequently impacting their capacity to interact with other proteins and facilitate membrane fusion [17]. Additionally, the functional attributes of SNARE proteins were influenced by the regulation of their phosphorylation status, protease activity under stress conditions [18]. *SYP21* and *SYP111*, two SNARE proteins in *Arabidopsis thaliana* (*A. thaliana*), play crucial roles in cytoplasmic division and salt stress response, respectively [19,20]. The *AtSYP4* protein family, comprising *AtSYP41*, *AtSYP42*, and *AtSYP43*, exhibited a positive response to salt and osmotic stress [21]. The *OsSYP71* overexpression enhanced *Oryza sativa* resistance to oxidative stress, suggesting that SNARE proteins play a significant role in plant responses to abiotic stresses [22]. In conclusion, SNARE proteins were instrumental in conferring plant resistance to abiotic stress, with the stress environment exerting multifaceted effects on both *SNARE* gene expression and protein functionality. These findings pave the way for a more in-depth exploration of the mechanisms underlying plant resistance to abiotic stress.

Genomic sequence analysis and comparative studies have shown that there are 36 SNAREs in the liverwort *Marchantia polymorpha* (*M. polymorpha*), 64 in (*A. thaliana*), and 60 in *Oryza sativa* (*O. sativa*) [23,24,25], a number significantly higher than that found in yeast and mammals [26,27]. These findings highlight the diversity and importance of SNARE proteins in plants, indicating their role in the development of a complex endomembrane system that may be associated with various living conditions and intricate developmental processes. The complexity of the endomembrane system necessitates the involvement and regulation of a greater number of SNARE proteins, culminating in the establishment of an extensive *SNARE* family with crucial biological functions in plant growth and development [28]. The SNARE gene family has been identified in a diverse range of plants, such as *O. sativa*, *A. thaliana*, *Zea mays* (*Z. mays*) [13], and *Glycine max* (*G. max*) [29]. This suggests that SNARE proteins are widely distributed and play a pivotal role in these plants.

Grapes, a popular fruit, often encounter extreme environmental stresses. For example, low temperatures and salinity have a profound effect on their yield and quality [30]. The SNARE gene family has been identified in a variety of plants, but the number of members of this family and the expression and function of each member under different stresses have not been reported in grapevine. Based on this, in this study, the physicochemical properties, evolutionary relationships, gene structure, codon preference, predicted protein interaction, *cis*-acting elements, and tissue-specific expression patterns of the grapevine *SNARE* gene are investigated using bioinformatics approaches. Quantitative real-time PCR (qRT-PCR) was used to assess the expression level of *SNARE* gene under 4 °C, ABA, NaCl, and PEG treatments. Finally, the analysis of subcellular localization of *VvSNARE37*, *VvSNARE44*, and *VvSNARE46* was located in a model system using the *Nicotiana benthamiana* system. In summary, this provides new insights and ideas for further study of *SNARE* gene expression under various stresses and subsequent development of grape varieties with greater tolerance to environmental stresses.

## 2. Results

### 2.1. Identification and Physicochemical Property Analysis SNARE Gene Family in Grapevine

In total, 52 grapevine *SNARE* genes were identified, sourced from the Plant Genome Database, and predominantly distributed on 18 chromosomes. These were designated *VvSNARE1*–*VvSNARE52* according to their chromosomal locations (Appendix A). Chromosomal localization revealed that the *VvSNARE* genes were distributed on eight chromosomes. Notably, Chromosome 5 harbored the highest number of *VvSNARE* genes, including *VvSNARE15*, *VvSNARE17*, *VvSNARE14*, *VvSNARE16*, *VvSNARE12*, and *VvSNARE13*. Chromosomes 4, 7, 8, and 12 contained four *VvSNARE* genes. Chromosomes 3, 9, 17, and 19 contained only one *VvSNARE* gene. Among them, the amino acid sequence of *VvSNARE41* was the shortest, with 95 bp. That of *VvSNARE14*, extending to 439 bp, was the longest. The molecular weights ranged from 10,904.83 (*VvSNARE41*) to 4,870,800.86 Da (*VvSNARE14*). Isoelectric points ranged from 4.95 (*VvSNARE13*) to 9.61 (*VvSNARE46*), as detailed in Appendix A.

### 2.2. Evolutionary Tree, Secondary Structure, and Subcellular Localization of the VvSNARE Gene Family in Grapevine

A phylogenetic tree was developed based on the amino acid sequences of the SNARE protein from both *A. thaliana* and grapevine (Figure 1). The SNARE gene family was categorized into 4 subfamilies, comprising 22 members in Subfamily I, 23 in Subfamily II, 21 in Subfamily III, and 23 in Subfamily IV. Notably, subfamily III contained the greatest number of members. Secondary structure prediction (Appendix A) showed that SNARE gene family predominantly consisted of α-helices (35.71–86.88%), irregular random coils (9.05–38.17%), and a lesser proportion of β-turns (0.71–9.61%). Subcellular localization prediction indicated that the *VvSNARE* gene family localized in the cytoplasm, nucleus, plasma membrane, Golgi apparatus, and vesicles (Appendix A).

### 2.3. Gene Structure, Motif, and Promoter Cis-Acting Elements

Examination of gene structures revealed a range of exons from 1 to 13 (Figure 2). Notably, *VvSNARE24* and *VvSNARE34* exhibited the highest number of exons, both 13, while *VvSNARE7*, *VvSNARE8*, *VvSNARE26*, *VvSNARE32*, *VvSNARE41*, and *VvSNARE51* had the lowest exon counts, all only 1. In order to gain a deeper understanding of the evolutionary and structural diversity within *VvSNARE*. The analysis of the conserved motifs in VvSNARE proteins were conducted utilizing the MEME online software. A total of 10 discrete and highly conserved motifs were identified (Appendix A). The findings revealed that the motif distribution pattern in the majority of VvSNARE proteins exhibited a high degree of conservation. However, Motif 9 exclusively existed in *VvSNARE23*, Motif 2 exclusively existed in *VvSNARE45*, and both *VvSNARE43* and *VvSNARE44* exclusively featured Motif 1.

In order to investigate the mechanisms of *VvSNARE* genes involved in plant growth, development, and responses to environmental factors, an analysis of *cis*-regulatory elements was conducted for the upstream 2000 bp sequences of the *VvSNARE* gene promoter using PlantCARE (Figure 3). The findings revealed that the *cis*-acting elements were categorized into four types: light-responsive elements, hormone-responsive elements, abiotic stress-responsive elements, and growth- and development-related elements. Among them, the hormone-responsive elements were related to abscisic acid, methyl jasmonate, gibberellin, and salicylic acid. Additionally, abiotic stress elements were associated with low temperatures and drought responses. The predicted results showed that the *SNARE* gene was responsive to a wide range of exogenous hormones and abiotic stresses of environmental factors. This suggested that *SNARE* genes might serve significant regulatory functions in response to exogenous hormones and abiotic stresses.

### 2.4. Collinearity Analysis, Codon Preference, and Selection Pressure Analysis

In order to gain deeper insights into the phylogenetic mechanisms and homology relationships of *SNARE* genes in grapevines. A comprehensive synteny analysis was conducted to compare grapevine with three representative species: *A. thaliana*, *Malus domestica* (*M*. *domestica*), and *O. sativa*. The internal collinearity analysis of the *VvSNARE* genome revealed a total of 10 pairs of duplicated gene pairs (Figure 4A), namely *VvSNARE3*/*VvSNARE40*, *VvSNARE3*/*VvSNARE46*, *VvSNARE30*/*VvSNARE11*, *VvSNARE29*/*VvSNARE9*, *VvSNARE39*/*VvSNARE22*, *VvSNARE42*/*VvSNARE5*, *VvSNARE44*/*VvSNARE4*, *VvSNARE7*/*VvSNARE10*, *VvSNARE17*/*VvSNARE21,* and *VvSNARE18*/*VvSNARE25*. *VvSNARE3* has two tandem repeats, and these results suggested that some *VvSNARE* genes may arise through gene duplication, which may have similar functions. The findings revealed the presence of 26 pairs of homologous genes between grapevines and *A. thaliana*, with a notable concentration in grapevine chromosomes chr2, chr3, chr4, chr5, chr6, chr7, chr8, chr11, chr12, and chr16. Remarkably, 66 pairs of homologous genes were identified between *Vitis vinifera* (*V. vinifera*) and *M. domestica*, with a significant concentration in grapevine chromosomes chr1, chr2, chr3, chr4, chr5, chr6, chr7, chr8, chr11, chr12, chr13, chr16, chr18, chr18-random, and chrUn. The five pairs of homologous genes were identified between grapevines and *O. sativa*, primarily concentrated in grapevine chromosomes chr1, chr2, and chr11, with 1, 1, and 3 pairs of homologous genes, respectively. This observation suggested that grapevine and dicotyledonous plants had more homologous genes than monocotyledonous plants (Figure 4B).

The average GC and GC3s content of the protein coding sequences of the *SNAREs* were 0.45 and 0.44, respectively, which suggests that these 52 genes exhibited a bias towards A and T bases across the entire coding frame region. Notably, 14 genes displayed positive Gravy values, signifying that these members were characterized as non-hydrophilic proteins, while the remaining genes exhibited hydrophilic properties. The analysis of effective codon count results indicated that the Nc values of all genes, with the exception of *VvSNARE26*, *VvSNARE41*, and *VvSNARE45*, exceeded 45. Furthermore, the CAI values for the *VvSNARE* genes ranged from 0.16 to 0.28, suggesting that the codon preference for these genes was relatively weak. The analysis of relative synonymous codon usage (RSCU) revealed that the *VvSNARE* gene exhibited preferences for specific codons, including UUG, CUU, CUG, AUU, ACG, GUU, GCU, GCA, CAA, CAG, AAU, AAC, AAA, AAG, GAU, GAC, CAA, GAG, AGA, and GGA (Figure 5A).

The Ka/Ks allowed estimation of their evolutionary selection pressure to further understand the evolutionary relationships of the grape SNARE gene family. From 10 pairs of genes with collinear relationships, the Ka/Ks values of six pairs of genes were calculated to be less than 1, suggesting that the grape SNARE gene family may be dominated by purifying selection. (Appendix A).

### 2.5. Expression Pattern Analysis and Protein Interaction Prediction

An analysis of the expression patterns of the *VvSNARE* gene family was conducted. Expression profiling revealed that a significant proportion (60%) of *SNARE* genes exhibited high expression levels in flesh and pollen. The remaining 41 *VvSNARE* genes exhibited lower expression levels in leaves, with the exception of *VvSNARE3*, *VvSNARE44*, *VvSNARE30*, *VvSNARE23*, *VvSNARE48*, *VvSNARE37*, *VvSNARE33*, *VvSNARE41*, *VvSNARE45*, *VvSNARE43*, and *VvSNARE26*. Notably, *VvSNARE18*, *VvSNARE32*, and *VvSNARE7* exhibited exclusive high expression levels in pollen. *VvSNARE11* showed high expression level in flesh, pericarp, pollen, and tendrils. High expressions of *VvSNARE12* and *VvSNARE16* were observed in seeds. *VvSNARE30* had high expression levels in leaf tissues. Pollen exhibited high expression of *VvSNARE49*. In summary, the *VvSNARE* gene family displayed high expression levels in flesh, pollen, and pericarp while exhibiting lower expression in leaves (Figure 5B).

Interactions among the 52 VvSNARE proteins were predicted using the STRING online platform (Appendix A). The findings revealed that 16 out of the 52 VvSNARE proteins collectively formed complex network structure with D7U2W9_VITVI (a protein containing the t-SNARE coiled–coil homology domain) and D7TSA1_VITVI (a protein containing the Vps53_N domain). Among them, VvSNARE27, VvSNARE10, and VvSNARE38 proteins were found to interact with D7TSA1_VITVI. Furthermore, VvSNARE10, VvSNARE38, VvSNARE27, and VvSNARE43 proteins displayed interactions with D7U2W9_VITVI. Extensive interaction existed between the family members. In addition, VvSNARE22 and VvSNARE39 also formed an interactive network with F6I3H9_VITVI and Vps53_N domain-containing protein. Moreover, both VvSNARE22 and VvSNARE39 interact with both F6I3H9_VITVI and Vps53_N domain-containing protein.

### 2.6. Quantitative Real-Time Fluorescence Analysis of the SNARE Genes in Grapevine

Quantitative real-time fluorescence analysis of *SNARE* genes (Figure 6A,B) revealed notable up-regulation of *VvSNARE2*, *VvSNARE14*, *VvSNARE15*, and *VvSNARE17* in response to ABA treatment. In contrast, *VvSNARE2*, *VvSNARE15*, *VvSNARE18*, *VvSNARE19*, *VvSNARE20*, *VvSNARE24*, *VvSNARE25*, and *VvSNARE29* genes displayed their highest expressions under NaCl treatment. PEG treatment led to significant down-regulation of *VvSNARE5*, *VvSNARE8*, *VvSNARE23*, *VvSNARE25*, *VvSNARE26*, *VvSNARE31*, and *VvSNARE49* genes. While at 4 °C, the expression of *VvSNARE37*, *VvSNARE39*, *VvSNARE44*, *VvSNARE45*, and *VvSNARE46* genes was significantly elevated compared to the control. These results indicated that different members of the *SNARE* gene family functioned under different stress treatments.

### 2.7. Subcellular Localization Assay of the SNARE in Grapevine

Based on qRT-PCR analysis, we observed *VvSNARE* gene family with a broad range of adaptability to plant stress, and most of *VvSNARE* genes were significantly induced by low temperature. The*VvSNARE37*, *VvSNARE44*, and *VvSNARE46* were selected for subcellular localization studies. In this investigation, we introduced 35S::*VvSNARE37*::EGFP, 35S::*VvSNARE44*::EGFP, and 35S::*VvSNARE46*::EGFP constructs into the subepidermal cells of tobacco leaves method and followed with transient expression analysis. The results revealed that green fluorescence was observed in the cell nucleus, cell membrane, and cytoplasm of cells injected with the empty 35S::EGFP construct, and similar green fluorescence patterns were observed in tobacco leaves injected with 35S::*VvSNARE37*::EGFP, 35S::*VvSNARE44*::EGFP, and 35S::*VvSNARE46*::EGFP constructs. This experiment confirmed that the *VvSNARE37*, *VvSNARE44*, and *VvSNARE46* genes were localized at the cell membrane, aligning with the subcellular localization predictions (Figure 7).

## 3. Discussion

Plant cells accomplish material transport through a complex endomembrane system and vesicle trafficking mechanism. In these processes, various factors play regulatory roles, such as coat proteins, Rab proteins, and SNARE proteins [31]. The primary function of SNARE proteins is to facilitate fusion between intracellular vesicles and the cell membrane [32]. Previous studies, through sequence analysis and comparison of the genomes of *A. thaliana* and *O. sativa*, have revealed that the *A. thaliana* genome contains 64 SNARE members, while the rice genome contains 60 [13]. This study comprehensively identified the *VvSNARE* gene family and discovered 52 *VvSNARE* family members. These members are widely distributed over 20 chromosomes of grapevine, with the highest number of genes located on Chromosome 5. It is worth noting that there is a significant difference in the number of SNARE gene family members between *A. thaliana*, *O. sativa*, and *V. vinifera*. This difference may be related to variations in the mechanisms of material transport during growth, development, and cellular differentiation processes.

The subcellular localization of genes within cells significantly influences their functionality [33]. In previous study, it has been found that in *A. thaliana*, SNARE proteins are confined to the cell membranes, including the plasma membrane, endoplasmic reticulum membrane, and Golgi apparatus membrane [34]. Conversely, in *O. sativa*, they have been found to exist on membranes of multiple organelles, such as the plasma membrane, endoplasmic reticulum membrane, Golgi apparatus membrane, vesicle membrane, and cytoplasmic membrane [35]. The subcellular localization prediction revealed that the prevalent presence of SNARE genes in the cell membranes of grapevine cells, consistent with previous research findings. Furthermore, the tobacco transient transformation experiments confirmed the cell membrane localization of the *VvSNARE37*, *VvSNARE44*, and *VvSNARE46* genes, validating the earlier predictions. However, there are differences in the subcellular distribution of SNARE gene family members between different species, possibly due to variations in growth and developmental conditions between species.

Protein–protein interactions play a pivotal role in modulating cellular functions, signal transduction, and metabolic pathways [36]. Studies have revealed that *A. thaliana* SNARE proteins interact with the following proteins: VAMP721 (Vesicle-Associated Membrane Protein 721), SNAP33, SNAP23, SYP121 (Syntaxin of Plants 121), and SYP61 (Syntaxin of Plants 61) [37]. Similarly, the investigations have identified interacting proteins for *Z. mays SNARE* genes, including SYP121 (Synaptobrevin-like protein), SYP122 (Synaptobrevin-like protein), SYP131 (Synaptobrevin-like protein), VAMP72 (Vesicle-associated membrane protein 72), and SNAP33 (Soluble NSF-attachment protein 33) [38]. In the case of soybean, SNARE proteins interact with VAMP (Vesicle-Associated Membrane Protein), syntaxin, and SNAP (Soluble NSF Attachment Protein) [39]. These proteins are tasked with facilitating intracellular transport and substance exchange through the formation of SNARE complexes, which mediate the fusion of intracellular membrane vesicles with the plasma membrane. They assume pivotal roles in the growth and development of *A. thaliana*, *Z. mays*, and *G. max*, in addition to participating in a multitude of physiological processes within cells. In this study, 16 out of the 52 VvSNARE proteins collectively formed complex network structure with D7U2W9_VITVI (a protein containing the t-SNARE coiled–coil homology domain) and D7TSA1_VITVI (a protein containing the Vps53_N domain). Among them, *VvSNARE27*, *VvSNARE10*, and *VvSNARE38* were found to interact with D7TSA1_VITVI. Furthermore, *VvSNARE10*, *VvSNARE38*, *VvSNARE27*, and *VvSNARE43* displayed interactions with D7U2W9_VITVI. Extensive interaction existed among the family members. In addition, *VvSNARE22* and *VvSNARE39* also formed an interactive network with F6I3H9_VITVI- and Vps53_N-domain-containing proteins. Moreover, both *VvSNARE22* and *VvSNARE39* interact with both F6I3H9_VITVI- and Vps53_N-domain-containing proteins. These interactions imply that VvSNARE gene family proteins may wield significant influence over grapevine growth and development.

In the evolutionary process of plants, they have evolved various mechanisms to adapt to changes in the external environment, including strategies to cope with low temperatures, drought, and UV radiation [40]. Studies have shown that under oxidative stress conditions, such as H_2_O_2_-induced oxidative stress, the expression of *NPSN11* in rice is significantly upregulated, while under salt stress or osmotic stress conditions, the expression of *NPSN11* is significantly downregulated [41]. Conversely, research on mutants of the *AtVAMP71* gene has shown a significant increase in plant salt tolerance. Under salt stress conditions, wild-type plants suffer cell damage due to the presence of hydrogen peroxide-containing vesicles, while plants with a loss of *AtVAMP71* function selectively inhibit this vesicle fusion process, thereby alleviating salt stress damage. Furthermore, Seifikalhor M et al. elucidated the regulation of K^+^ and Cl^−^ channels via the plant ABA signaling pathway, indirectly integrating the influence of vesicle proteins into the regulation of the Ca^2+^ signaling pathway [42]. In addition, studies have identified the crucial role of the vesicle gene family in the adaptive responses of rapeseed to adverse conditions. It is noteworthy that specific vesicle genes are significantly upregulated in rapeseed when facing drought and salt stress, providing additional evidence for their involvement in stress responses [43].

The investigation unveiled that, upon examination via qRT-PCR, the expression of *VvSNARE2*, *VvSNARE14*, *VvSNARE15,* and *VvSNARE17* exhibited substantial upregulation in response to 0.2 mmol·L^−1^ ABA treatment when juxtaposed with the control. These findings strongly implicate the involvement of these genes in responding to ABA hormone treatment. Furthermore, the expression levels of *VvSNARE2*, *VvSNARE15*, *VvSNARE18*, *VvSNARE19*, *VvSNARE20*, *VvSNARE24*, *VvSNARE25,* and *VvSNARE29* underwent marked upregulation in response to 400 mmol·L^−1^ NaCl treatment. This implies their potential responsiveness to salt-induced stress. Moreover, under 10% PEG treatment, the expression levels of *VvSNARE5*, *VvSNARE8*, *VvSNARE23*, *VvSNARE25*, *VvSNARE26*, *VvSNARE31,* and *VvSNARE49* experienced substantial down-regulated in contrast to the control. Additionally, the expression levels of *VvSNARE37*, *VvSNARE39*, *VvSNARE44*, *VvSNARE45*, and *VvSNARE46* underwent significant upregulation at 4 °C. Taken together, these findings imply that *VvSNARE* genes may play a crucial role in response to high salt concentrations, low temperatures, drought-induced stresses, and exogenous hormone treatments. This reveals the complexity of VvSNARE genes in regulation and provides a solid foundation for further studies on the function of grape SNARE genes. However, although the above results provide some preliminary clues, the specific function of the VvSNARE gene still needs to be further verified by gene cloning and functional validation experiments.

## 4. Materials and Methods

### 4.1. Plant Materials and Treatment

Test tube plantlets of “Pinot Noir” (*Vitis vinifera* L.) were utilized experimental specimens in this study. The test tube plantlets were meticulously cultivated and preserved in the Laboratory of Fruit Tree Physiology and Biotechnology at Gansu Agricultural University. Single-bud stem segments were subjected to grafting onto 50 mL of GS solid medium, followed by subculturing under white LED illumination conditions (16 h of light at 25 °C and 8 h of darkness at 20 °C) for a duration of 30 days. Test tube plantlets exhibited robust growth and were free from any signs of contamination were chosen for further experimentation. The roots were meticulously excised from the culture medium, and an extensive cleansing procedure with sterilized water was carried out on a highly controlled, super-clean workstation. Subsequently, the plantlets were subjected to additional cultivation in GS liquid medium supplemented with 400 mmol·L^−1^ NaCl, 10% PEG, and 0.2 mmol·L^−1^ ABA, while a control group received an equivalent volume of distilled water treatment. Low-temperature stress conditions at 4 °C were imposed within a cryogenic plant incubator, and normally developed test tube plants were employed as the control group. The entire duration of each treatment was set at 24 h, and every treatment was replicated in triplicate. Following the treatments, samples were collected, Liquid nitrogen preservation, and RNA extraction was performed.

### 4.2. Identification of the SNARE Gene Family in Grapevines

Protein sequences of SNARE genes were obtained from the A. thaliana database (https://www.arabidopsis.org/, accessed on 28 March 2023). Grapevine genome and annotation information was downloaded from phytozome v13 (https://www.arabidopsis.org/, accessed on 23 May 2024). The grape genome and annotation information were downloaded from phytozome v13 (https://phytozome.jgi.doe.gov/pz/portal.htm, accessed on 30 March 2023). All protein sequences of grape were extracted and compared with AtSNARE family protein sequences using the TBtools software (version 1.108) to obtain preliminary grape SNARE family members. Then, we compared these preliminary screened protein sequences using NCBI’s Protein Blast panel on the Plant Genome website (https://phytozome.jgi.doe.gov/pz/portal.html, accessed on 15 April 2023) to obtain the accession number, full length of the gene, CDS sequence, and protein sequence of the grape SNARE gene. The sequences were screened for structural domains using HMMER (https://www.ebi.ac.uk/Tools/hmmer/, accessed on 20 April 2023) to remove sequences that did not contain SNARE-specific structural domains. Physical and chemical properties were analyzed using the online software ExPASy (https://web.expasy.org/protparam/, accessed on 22 April 2023).

### 4.3. Phylogenetic Evolution, Secondary Structure and Subcellular Localization

The multiple sequence alignment of the VvSNARE proteins were conducted using the ClustalX 1.83 software. In MEGA 7.0 software, the neighbor-joining method (NJ) was used to construct an evolutionary tree, and the bootstrap value was 1000, using EVOLVIEW website (https://evolgenius.info//evolview-v2/#login, accessed on 24 April 2023) for beautification. The NPS@: SOPMA website (https://npsa-prabi.ibcp.fr/cgi-bin/npsa_automat.pl?page=npsa_sopma.html, accessed on 26 April 2023) was used to predict the secondary structures of VvSNARE proteins. The online software WoLF PSORT (https://wolfpsort.hgc.jp/, accessed on 28 April 2023) was used to predict the subcellular localization of the VvSNARE proteins.

### 4.4. Analysis of Gene Structure, Motif, Domain, and Cis-Acting Elements

Gene structure analysis was performed using the online software GSDS2.0 (http://gsds.gao-lab.org/index.php, accessed on 8 May 2023). The conserved motifs of proteins were constructed by the MEME (http://meme-suite.org/tools/meme, accessed on 12 May 2023), the number of motifs was set to 10, and the remaining parameters were all default values. The conserved domains of the protein were analyzed at theNCBI-CDD website (https://www.ncbi.nlm.nih.gov/cdd/, accessed on 18 May 2023). The sequence 2000 bp upstream of transcription start site (TSS) of each *VvSNARE* was extracted from the phytozome database and analyzed using the online software New PLACE (https://www.dna.affrc.go.jp/PLACE/?action=newplace, accessed on 20 May 2023) and mapped in TBtools (version 1.108).

### 4.5. Gene Location and Synteny Analysis

Chromosome localization of grapevine SNARE gene family members was performed using the TBtools (Version 1.108) software. To analyze the collinearity relationships of VvSNARE genes, the genome and annotation files of *A. thaliana*, *M. domestica*, *V. vinifera,* and *O. sativa* used for collinearity analysis were downloaded from phytozome v13 (https://phytozome.jgi.doe.gov/pz/portal.html, accessed on 25 May 2023), the gene pairs of the *VvSNARE* genes were determined using TBtools synteny, and the diagram was drawn via TBtools (Version 1.108).

### 4.6. Codon Bias and Selective Pressure Analysis

The codon usage characteristics of the CDS sequence of *VvSNARE* genes were analyzed using the online software CodonW 1.4.2 (http://codonw.sourceforge.net, accessed on 27 May 2023). The relative synonymous codon usage (RSCU), effective codon (ENC), codon bias index (CBI), codon adaptation index (CAI), optimal codon usage frequency (Fop), T3s, C3s, A3s, G3s, With T3s, C3s, A3s, G3s, CAI, CBI, Nc, Fop, GC, GC3s, L_sym, L_aa, GRAVY, and Aromo parameter correlation analysis were analyzed. Using TBtools’ NGmethod, Ka (nonsynonymous replacement rate), Ks (synonymous replacement rate), and Ka/Ks (selection intensity) were calculated for 10 pairs of *VvSNARE* genes with collinear relationships.

### 4.7. Expression Pattern and Protein Interaction Analysis of SNARE Gene Family in Grapevines

The expression levels of *VvSNARE* genes in different tissues of grapevine, including flesh, pericarp, seed, pollen, petal, stamen, carpel, root, flower, bud, branch, tendril, and leaf, were searched in the BAR database (https://bar.utoronto, accessed on 28 May 2023) and plotted in TBtools for graphing. The protein interaction network was predicted by STRING Version 11 (https://string-db.org/, accessed on 29 May 2023).

### 4.8. qRT-PCR Analysis

The primers were synthesized by Shanghai (Shanghai, China) Biological Engineering Co., Ltd. (Appendix A). The RNA from grapevine leaves was extracted by the CTAB method, and the Evo M-MLV RT Kit with Gdna Clean for qPCR II Kit (TaKaRa Biotechnology, Lanzhou, China) was reverse transcribed into cDNA and diluted to the same concentration as a real-time fluorescent quantitative PCR template. The total volume of the qRT-PCR reaction system was 20 μL, including 10 μL of 2 × SYBR, 1 μL of cDNA, 1 μL each of upstream and downstream primers, and 7 μL of ddH_2_O.

### 4.9. Cloning of the VvSNARE Gene and Transient Transformation of Nicotiana Benthamiana

In this study, *VvSNARE37* (*VIT_213s0101g00410*), *VvSNARE44* (*VIT_216s0039g01850*), and *VvSNARE46* (*VIT_217s0000g06580*) were screened, which were significant in responding to low-temperature stress by analyzing the quantitative data of the gene families, and synthesized the primers with specific EcoRI and XhoI cleavage sites in CE Design V1.04 using SnapGene software (7.0) to design specific primers with EcoRI and XhoI cleavage sites and synthesized in CE Design V1.04 (Appendix A). PCR amplification was performed using “Pinot Noir” cDNA as a template, and the recovered product was ligated with pART-CAM-EGFP plasmid. The ligated product was then transferred into *DH5α* coloreceptor cells, which were detected by bacteriophage PCR and then sent to Sangon Biotech (Shanghai, China) Co., Ltd. for sequencing. After appropriate sequencing, the plasmid was extracted to transform *Agrobacterium tumefaciens,* and the positive was identified after sequencing. The plasmid was extracted and transformed into *Agrobacterium tumefaciens*, and the positive *Agrobacterium tumefaciens* was identified for subsequent plant transformation experiments. PCR amplification was performed using “Pinot Noir” cDNA as template, and the obtained product was ligated with pART-CAM-EGFP plasmid, followed by transfection of the ligated product into *DH5α* coloreceptor cells, which was detected by bacteriophage PCR and then sent to Shanghai Shenggong Biological Engineering Co. for sequencing. After appropriate sequencing, the plasmid was extracted to transform *Agrobacterium tumefaciens*, and positive *Agrobacterium tumefaciens* was identified for subsequent plant transformation experiments.

### 4.10. Subcellular Localization of VvSNARE in the Lower Epidermis Cells of Nicotiana Benthamiana

The fusion plasmids were transferred into *Agrobacterium tumefaciens* strain GV3101 using the freeze–thaw method, and the bacteria solution was incubated in LB medium containing 50 μg·mL^−1^ spectacularinomycin and 50 μg·mL^−1^ rifampicin at 28 °C until the OD_600_ reached 0.6. The solution was centrifugated at 8000 rpm for 5 min at 4 °C. The agrobacteria were resuspended using infiltration medium containing 1 M MES (pH = 5.6), 2.5 M MgCl_2_, and 100 mM acetosyringone until the OD_600_ reached 0.75. After resuscitation for 3 h, the resuspension solution was injected into 4-week-old *Nicotiana benthamiana* leaves using a needleless syringe. The injected tobacco was cultured in dark for 24 h and then the fluorescence of EGFP was detected by confocal laser scanning microscope (Olympus FV1000 Viewer, Tokyo, Japan).

### 4.11. Statistical Analysis and Data Processing

The test data were statistically analyzed using Excel 2010 and SPSS 22.0 software. The data were significant (*p* < 0.05) using one-way ANOVO Duncan detection, and Origin Pro 2022 software was used for drawing.

## 5. Conclusions

In the present study, a total of 52 members were distributed across 20 distinct chromosomes. The results of qRT-PCR showed that *VvSNARE14* and *VvSNARE15* had higher expression levels under ABA treatment, and *VvSNARE18*, *VvSNARE19,* and *VvSNARE29* had higher expression levels under sodium chloride treatment. The expression levels of *VvSNARE1*, *VvSNARE26,* and *VvSNARE31* were significantly down-regulated under PEG treatment, while the expression levels of *VvSNARE37*, *VvSNARE44,* and *VvSNARE46* were significantly up-regulated under 4 °C treatment. These genes can be used as candidate genes for further functional studies. Our transient expression experiments in tobacco leaves showed that *VvSNARE* genes are mainly localized in cell membranes. In conclusion, this study provides a new way to further study the multi-role of VvSNARE gene family in coping with various abiotic stresses.

## Figures and Tables

**Figure 1 ijms-25-05984-f001:**
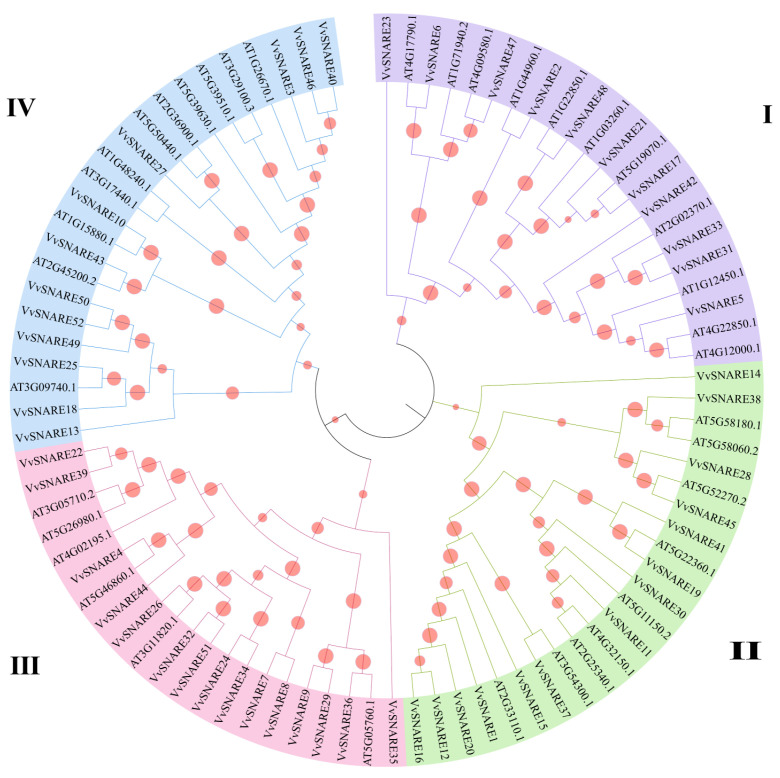
Phylogenetic analysis of the SNARE gene family in grapevine. Phylogenetic trees were constructed using the SNARE protein sequences. I–IV represent different subgroups, respectively. The NJ method was used, and the bootstrap value was set to 1000.

**Figure 2 ijms-25-05984-f002:**
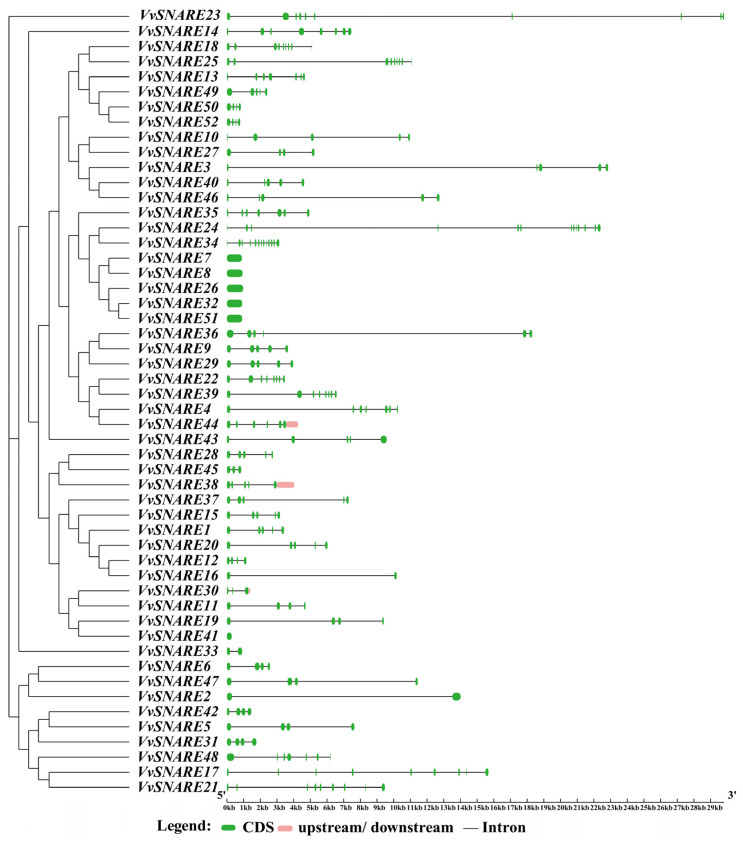
Gene structure analysis of SNARE gene family in grapevine. The exon–intron structure of *VvSNARE* genes. Exon CDS, upstream/downstream, and intron are indicated by the green boxes, pink boxes, and black lines, respectively. The scale bar represents 1 kb (right).

**Figure 3 ijms-25-05984-f003:**
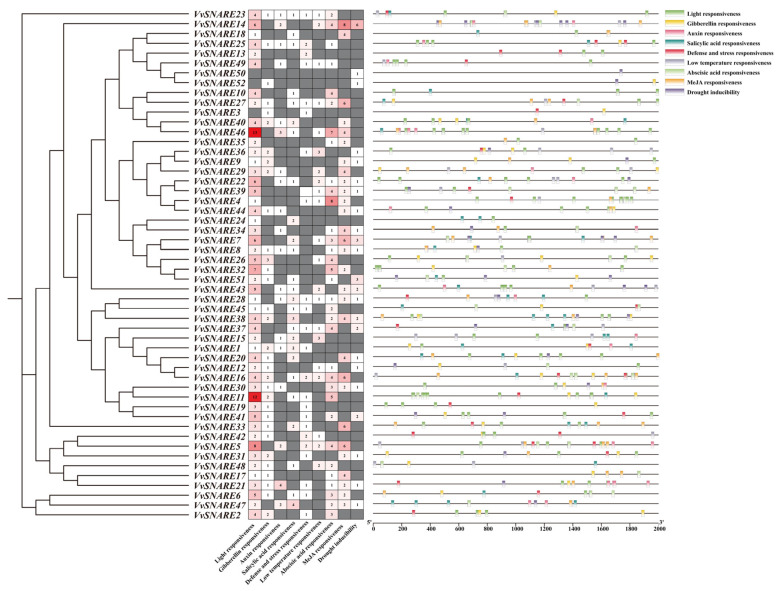
*Cis*-regulatory element analysis of the VvSNARE genes. Different colors on the right represent different elements. the numbers in the figure represent the number of cis acting elements.

**Figure 4 ijms-25-05984-f004:**
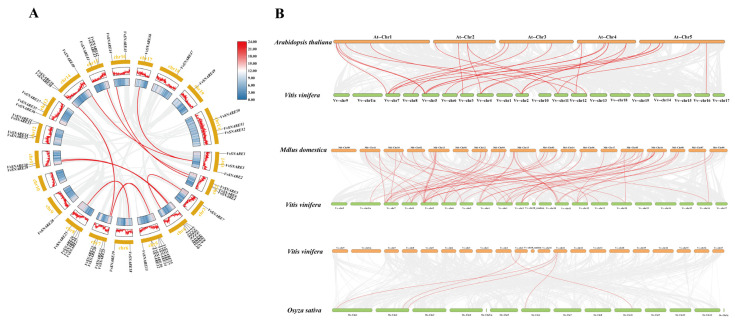
(**A**,**B**) Analysis of the collinear of the SNARE gene family in grapevines. (**A**) Collinearity analysis of *VvSNARE*. The gray lines represent all collinear blocks in the grape genome, and the red lines represent gene pairs between the *VvSNARE* genes. (**B**) Collinearity analysis of *SNARE* genes in grapevines and three representative plants. The gray lines in the background show collinearity between the *V. vinifera* and *A. thaliana*, *M. domestica,* and *O. sativa* genomes. The red lines show collinearity between *A. thaliana* and *V. vinifera*, *M. domestica* and *V. vinifera*, and *Oryza sativa* and *V. vinifera* for *VvSNARE* genes.

**Figure 5 ijms-25-05984-f005:**
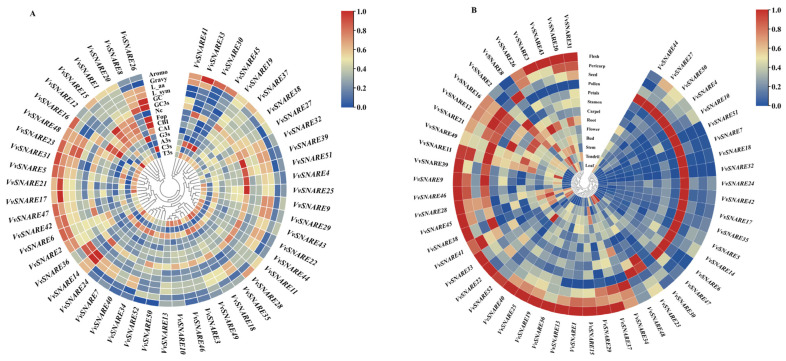
(**A**) Codon parameter analysis of the *SNARE* genes in grapevines. “A3s, G3s, C3s and T3s” refers to the synonymous codon corresponding base frequency on the third. “CAI” refers to the codon adaptation index, “CBI” refers to the codon bias index, “FOP” refers to the frequency of optimal codons, “ENc” refers to the effective number of codons, “GC3s” refers to the amount of the third codon (G + C), and “GC” refers to the count of genes (G + C). (**B**) Tissue differential expression of *VvSNARE*. Heatmap experiments were performed with GeneChip microarrays obtained from the grape tissue expression database. Red or blue shading represents the expression level—significant or insignificant, respectively. The scale indicates the relative expression level.

**Figure 6 ijms-25-05984-f006:**
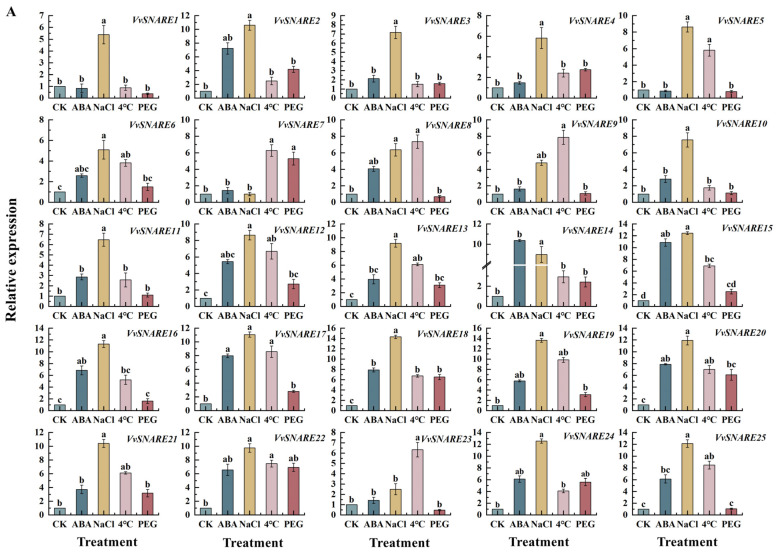
(**A**,**B**) Relative expression levels of SNARE genes in grape tissue under ABA, NaCl, 4 °C, and PEG treatments. Grape plantlets grown for 30 days were treated with 400 mmol·L^−1^ NaCl, 10% PEG and 0.2 mmol·L^−1^ ABA, respectively. The treatment time was 24 h. Distilled water was used as the control. Three biological and technical replicates were set up for this experiment. Gene expression was normalized to the unstressed control expression level, which was given the value of 1. Error bars represent the mean ± SE from three biological replicates. Different letters indicate significant differences, while the same lowercase letters indicate no statistical difference (*p* < 0.05).

**Figure 7 ijms-25-05984-f007:**
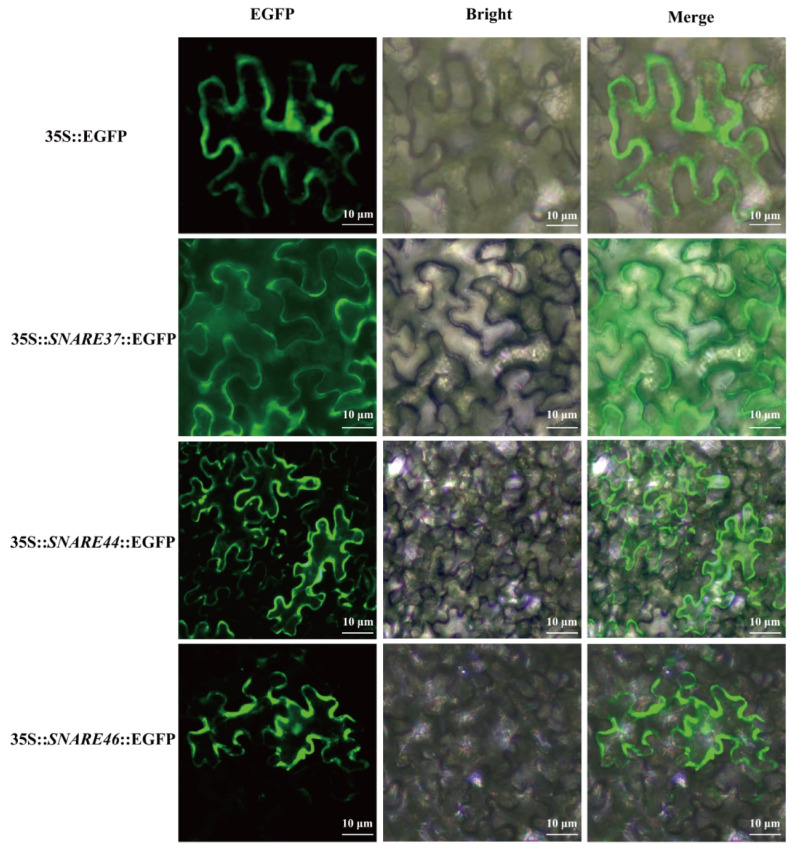
Subcellular localization of *VvSNAREs* in the lower epidermal cells of *Nicotiana benthamiana*. The 35S::*VvSNARE37*-EGFP, 35S::*VvSNARE44*-EGFP, and 35S::*VvSNARE46*-EGFP constructs are expressed in *Nicotiana benthamiana* leaves. The top row shows the positive control (35S::EGFP), and the bottom row shows the 35S::*VvSNARE*-EGFP construct. The EGFP fluorescence image, bright field (BF) image, and the merged EGFP and BF images are shown. Scale bars correspond to 10 µm.

## Data Availability

Data will be made available on request.

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
