# Peer review of "Molecular Evolution of SNAREs in *Vitis vinifera* and Expression Analysis under Phytohormones and Abiotic Stress"

_ijms, 2024, doi:10.3390/ijms25115984_

Round 1

Reviewer 1 Report

Comments and Suggestions for Authors

The manuscript, “Molecular evolution of…and abiotic stress,” authored by Zeng et al., explores the SNARE proteins in Vitis vinifera. In my opinion, the work provides some new insight into the distribution and evolution of SNARE genes in the grapevine genome. However, the manuscript is hastily prepared and lacks readability to the point where it becomes difficult to follow the presented data. I am detailing below a few shortcomings I encountered. The authors must address these before the article can be accepted.

1)    Important: Authors need to check references to make sure that they are not misquoted. I randomly checked reference 25 (quoted in lines 86 and 98); the original reference has explored Populus trichocarpa and not Populus nigra as quoted by the authors. Further, in line 98 authors suggest that reference 25 says the grapevine’s productivity is affected by abiotic stresses, but I could not find such a statement in the original reference.

2)    Introduction needs to be improved to provide adequate background while providing the gaps in the field.

3)    The discussion section is unnecessarily long. The authors should highlight important outcomes from the study and how the study improves the current understanding of the SNARE proteins. For example, in Lines 316 to 319, the authors have concluded that the SNARE proteins from grapevines predominantly function in the plasma membrane. How can such a generalized statement be based on the observation of three SNARE proteins when the total number of SNARE proteins is 52?

4)    The description of methods is adequate; however, in section 4.1, the authors state that the cut plantlets were treated for 24 hours in GS medium supplemented with ABA, NaCl, and PEG. In Figures 6 and 7, the authors present the individual influence of ABA, NaCl, and PEG on the expression of SNARE genes. Were the chemicals used in combination?

5)    Figures 6 and 7 have panels A and B, with panel A placed below panel B. Are they two individual figures, or is it Figure 6 with two panels?

6)    Table 1 is not in the manuscript, although the authors have referenced it two times (lines 217 and 489).

7)    Why are authors suddenly talking about the strawberry ANS gene family in lines 216 to 220?

Why is page 16 empty?

Comments on the Quality of English Language

Must be proofread to remove grammatical errors and spelling mistakes.

Reviewer 2 Report

Comments and Suggestions for Authors

The article "Molecular Evolution of SNAREs in Vitis vinifera and Expression Analysis under Phytohormones and Abiotic Stress" provides a detailed and balanced assessment of a study investigating the role of SNARE proteins in grapevine. The thorough identification and structural analysis of VvSNARE genes, as well as its insightful findings on promoter region enrichment and stress-responsive gene expression. However, it also highlights areas for improvement, including the need for further functional characterization beyond gene expression analysis, comparative functional analysis with other species, and expansion of subcellular localization assays to encompass more VvSNARE genes. Overall, the article acknowledges the study's significant contribution to understanding grapevine stress responses but suggests avenues for future research to strengthen its findings. However, some changes might required. The objective of the study needs to be more specified and elaborative at the end of the introduction section in a separate paragraph. The discussion looks fragmented, I recommend making a good story in the discussion including your objective and results. Take-home message is missing in the conclusion and abstract. The conclusion is lengthy, try to shorten it.
